# LLMZip: Lossless Text Compression using Large Language Models

## Abstract

We design a lossless compression algorithm for compressing English text by using the large language model LLaMA2-7B as a predictor for the next token given a window of past tokens. Specifically, the proposed LLMZip algorithm uses the conditional probabilities at the output of the large language model in conjunction with Arithmetic Coding. Our algorithm outperforms state-of-the-art text compression schemes such as BSC, ZPAQ, and paq8h. We show that it is possible to marginally improve the compression performance further by first extracting a summary from the document and compressing the text by conditioning on the summary. Finally, we investigate the compression performance of LLMZip when the summary (side information) is available both at the encoder and decoder. We show that the LLM is able to exploit the available side information to significantly improve the compression performance. As an important byproduct, we provide new estimates of an asymptotic upper bound on the entropy of English which is significantly smaller than currently available estimates.

## 1 Introduction

There are close connections between learning, prediction, and compression. The success of Chat-GPT has captured the fascination of the general public and brought the connection between learning and prediction to the fore. The main advance brought about by large language models such as LLaMA and GPT-4 is that they excel at predicting the next token in a text corpus, conditioned on knowing the past tokens within the context window.

The connection between prediction and compression was explored as early as 1951 by Shannon in order to estimate the entropy of the English language (Shannon (1951)). The idea that a good predictor for the $i$th value in a time series based on the past values can be effectively converted to a good compression algorithm has played a prominent role in information theory. Many algorithms for speech, image, and video compression exploit this notion, either explicitly or implicitly. The performance of such a compression scheme depends substantially on the efficacy of the predictor and every time there is a major advance in the prediction capability, it behooves us to study its effect on the compression performance. For lossless text compression, the idea of combining a language model with arithmetic coding has been shown to be effective (MacKay (2003)). Indeed, the authors in Goyal et al. (2018) use recurrent neural networks (RNN) as predictors, and they report improved results for certain types of source. Their scheme still did not outperform state-of-the-art algorithms such as BSC and ZPAQ for text compression.

Given the recent release of large language models such as the LLaMA models Touvron et al. (2023a),Touvron et al. (2023b), this is an opportune time to study whether one can obtain better compression results and sharper estimates of the entropy of the English language. This is the main goal of this paper.

### 1.1 Contributions

We study the compression performance of using a foundational model (LLaMA2-7B) in conjunction with entropy coding for lossless text compression, and the resulting suite of compression algorithms are broadly termed as LLMZip. We refer to the combination of using a large language model for predictive modeling followed by using arithmetic coding for entropy coding as the LLM+AC algo-

rithm. With the LLaMA2+AC algorithm, we obtain a compression ratio of 0.6936 bits/character on a 1MB section of the text8 dataset and a compression ratio of 0.7741 bits/character on a 200KB section of a text from a book that was released on Project Gutenberg after the release of LLaMA2 Cooke & Reed (2023). These compression ratios are significantly better than the compression ratio obtained using state-of-the-art text compressors such as BSC, ZPAQ and pq8h on the full 100MB of the text8 dataset. This paper is the first to demonstrate that excellent compression can be obtained using LLMs for compressing text that was not part of the training corpus.

We show that when the LLaMA2-7B large language model is used as the predictor, the asymptotic upper bound on the entropy is 0.692 bits/character when estimated using a 1MB section of the text8 dataset. This is smaller than earlier estimates provided in Cover & King (1978) and (Lutati et al., 2023, Table 4). The estimate of the upper bound increases to 0.77 bits/character for a 200 KB section of the text from Cooke & Reed (2023), which is still lower than the estimates in Lutati et al. (2023).

We introduce the idea of compressing text by first creating a summary and then conditioning on the summary to compress the text using the LLM+AC algorithm. We show that this provides marginal improvement to the compression ratio.

Finally, we study the compression performance of LLM+AC when side information such as a short summary of the text is present both at the encoder and decoder, and we show that the LLM is able to exploit the presence of side information to improve the compression performance significantly compared to when the side information is absent.

## 1.2 RELATED WORK

There is a rich body of literature that combines predictive modeling with entropy coding for obtaining lossless compression MacKay (2003). In particular, the use of predictive modeling with arithmetic coding dates back to the 1980s Cleary & Witten (1984); Witten et al. (1987); Willems et al. (1995). Neural networks such as fully connected neural networks and LSTMs have been used for predictive modeling, and they have been used in conjunction with arithmetic coding for lossless text compression in Mahoney (2000); Goyal et al. (2018). Using transformers for obtaining general purpose compression was considered in Mao et al. (2022),Bellard (2021). Recently, work that appeared soon after our report[1] Delétang et al. (2023) showed that a large language model Chinchilla can be used in conjunction with arithmetic coding to obtain similar compression performance to what is reported in this paper. They report the surprising result that a large language model trained on text data is also effective at compressing images and speech. Transformers have also been used in conjunction with arithmetic coding for compressing audio in Défossez et al. (2022).

## 2 PREDICTIVE MODELING AND COMPRESSION USING LLMS

Let $\mathbf{s}$ denote a sentence from the English language composed of $N_c$ letters, where each letter is assumed to be from the alphabet $\mathcal{S}$. We assume that we have a dictionary $\mathcal{X} = [1, D]$ of $D$ tokens. We first parse $\mathbf{s}$ into a sequence of $N_T$ tokens denoted by $\mathbf{x} = x_1, x_2, \ldots, x_{i-1}, x_i, x_{i+1}, \ldots x_{N_T}$, where $x_i \in \mathcal{X}$. There is a one-to-one mapping between $\mathbf{s}$ and $\mathbf{x}$ and hence, compressing $\mathbf{s}$ is the same as compressing $\mathbf{x}$. In this context, token $x_i$ can be thought of as realizations of the random variable denoted by the upper case letter $X_i$.

A language model with memory $M$ is a predictor that operates as follows. At epoch $i$, it accepts tokens $x_{i-M}, x_{i-M+1}, \ldots, x_{i-1}$ and produces a probability mass function (PMF) for the next token in the sequence, conditioned on the past $M$ tokens; we denote the ensuing conditional distribution by $q_i(x_i) := \Pr(X_i = x_i | x_{i-1}, x_{i-2}, \ldots, x_{i-M})$. The conditional PMF vector at epoch $i$, namely $\mathbf{q}_i$, is input to a lossless compression algorithm along with the actual sequence that needs to be compressed. The lossless compression algorithm produces a sequence of $N_b$ compressed bits. A schematic of this scheme is shown in Fig. 1.

The main metric of interest is the compression ratio $\rho$ defined as

$$\rho := \frac{N_b}{N_c} \text{bits/character.}$$

---

[1]We are not citing our own technical report due to double-blind requirements.

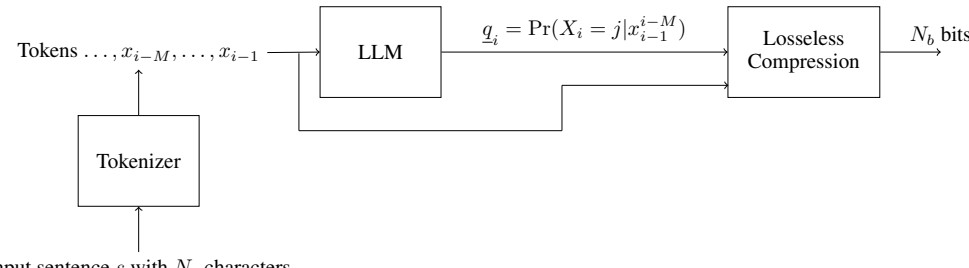

Figure 1: Schematic showing the prediction at epoch $i$.

## 2.1 ENCODING SCHEMES

We consider three schemes for the lossless compression block in Fig. 1.

### 2.1.1 COMPRESSING THE RANKS USING ZLIB

First, the PMF vector $\mathbf{q}_i := [q_i(1), q_i(2), \ldots, q_i(D)]^\mathsf{T}$ is sorted in descending order; we denote the sorted PMF vector by $\tilde{\mathbf{q}}_i$. Let $\gamma_i : \mathcal{X} \to \mathcal{X}$ be a permutation on the integers from 1 to $D$ such that

$$\tilde{q}_i(\gamma_i(j)) = q_i(j), \forall j \in \mathcal{X}.$$

That is, $\gamma_i(j)$ is the rank of the token $j$ at epoch $i$. We define the rank of the input sequence at epoch $i$ as the rank of the token $x_i$ at epoch $i$, $r_i := \gamma_i(x_i)$. The sequence $\{r_i\}_{i=1}^{N_T}$ is compressed by a lossless compression algorithm (such as zlib) to produce $N_b$ bits which are the final bit representation of the input sequence. The first scheme uses the zlib compression algorithm to encode the sequence of ranks. We refer to this scheme as LLaMA+zlib and denote the compression ratio of this scheme by $\rho_{\text{LLM+zlib}}$.

### 2.1.2 TOKEN-BY-TOKEN COMPRESSION

The second scheme uses a token-by-token lossless compression scheme which uses a time-varying codebook to encode the token $x_i$ at epoch $i$. The time-varying codebook is formed using a prefix-free code designed under the assumption that $q_i$ is the true token distribution. A natural choice for the prefix-free code is a Huffman code. Instead, for simplicity, we adopt a prefix-free code where the codeword for the token $x_i$ is of length $l_i = \lceil \log_2 \frac{1}{q_i(x_i)} \rceil$. A prefix-free code with this length for $x_i$ is guaranteed to exist since this choice of lengths satisfies the Kraft inequality Cover & Thomas (1999). The compression ratio for this scheme, denoted by $\rho_{\text{LLM+TbyT}}$, is given by

$$\rho_{\text{LLM+TbyT}} = \frac{\sum_{i=1}^{N_T} \left\lceil \log_2 \frac{1}{q_i(x_i)} \right\rceil}{\sum_{i=1}^{N_T} b_i}.$$

### 2.1.3 ARITHMETIC CODING

The above two schemes are intuitive, but their performance can be improved. A very effective way to combine the output of the LLM with a lossless compression scheme is by using arithmetic coding Rissanen & Langdon (1979); Bell et al. (1989); Cleary & Witten (1984). Arithmetic coding is well suited to accept time-varying probabilities, and we use $q_i(x_i)$ as the probability of token $x_i$ at epoch $i$ in the arithmetic coding scheme. Figure 2 shows a schematic of how the encoding procedure works when using Arithmetic Coding. We denote $\mathbf{Q}_i$ as the conditional cumulative mass function (CMF) vector at epoch $i$, with $Q_i(j) = \sum_{k=1}^{j} q_i(k)$. In Arithmetic Coding, an interval $\mathcal{B}_i := [B_{i,min}, B_{i,max})$ is maintained and updated at epoch $i$. We start with the initial condition $\mathcal{B}_0 = [0, 1)$ and during the $i$th epoch, we update $\mathcal{B}_i$ according to

$$B_{i,min} = B_{i-1,min} + (B_{i-1,max} - B_{i-1,min})Q_i(x_i - 1) \tag{1}$$

$$B_{i,max} = B_{i-1,min} + (B_{i-1,max} - B_{i-1,min})Q_i(x_i), \tag{2}$$

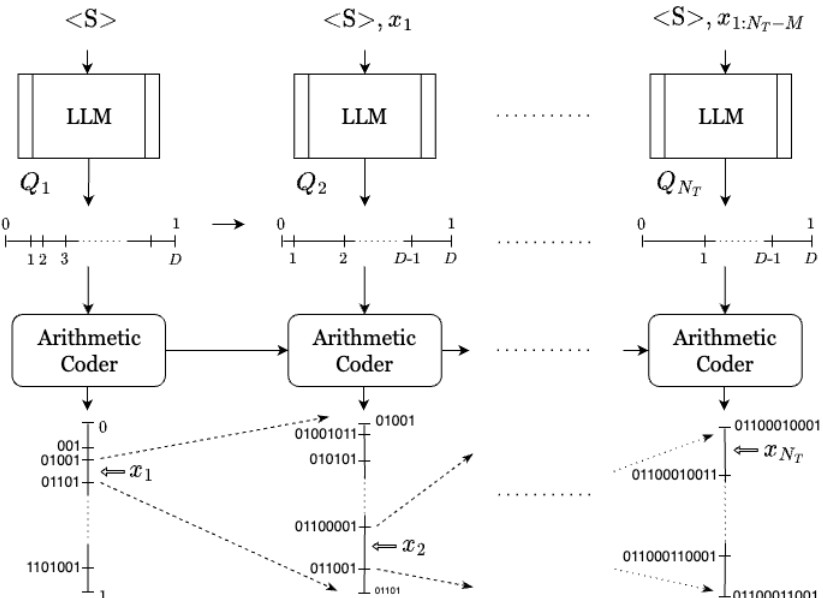

Figure 2: Schematic showing the working of LLM in conjunction with an Arithmetic coder for encoding a sequence of tokens in LLMZip.

where $x_i$ is the token in the input text at epoch $i$ that is to be encoded and $Q_i(0)$ is set to zero $\forall i$. At the end of the token sequence, any real number $C$ that lies in the interval $\mathcal{B}_{N_T}$ is selected and the binary representation of $C$ is selected as the compressed bit sequence corresponding to the sequence of input tokens. In the illustrative example shown in Fig. 2, the binary representations of $B_{i,min}$ and $B_{i,max}$ are shown in the figure. At the final stage the binary representations corresponding to $B_{N_T,min}$ and $B_{N_T,max}$ are 01100010001 and 01100010011, respectively and the final bit representation is 01100010010, which lies in $\mathcal{B}_{N_T}$. Note that in this example, encoding the first token fixes the first two bits of the final bit sequence, while encoding the second token fixes the next three bits.

The decoder also uses the same LLM (with the same parameters) as the encoder. Decoding begins by inputting the start token to the LLM at time $i = 1$ and computing $\mathbf{Q}_1$. Then, the decoded symbol $x_1$ is given by the $j$ for which $C \in [Q_1(j-1), Q_1(j))$. We maintain an interval $\mathcal{B}_1$ at the decoder and update it using equation 1 and equation 2. Then, we add $x_1$ to the context window and proceed to compute $\mathbf{Q}_2$. Thus, the decoder can mimic the same operations of the encoder to produce $Q_i$ during epoch $i$ and determine $x_i$ as the symbol $j$ for which $C \in [B_{i-1,min} + (B_{i-1,max} - B_{i-1,min})Q_i(j-1), B_{i-1,min} + (B_{i-1,max} - B_{i-1,min})Q_i(j))$. The decoder then updates $\mathcal{B}_i$. The decoder can also work in a streaming fashion, but we don't discuss this further.

We refer to the compression ratio of this scheme as $\rho_{\text{LLM+AC}}$. It is known that arithmetic coding is nearly optimal as a compression scheme (MacKay, 2003, Page 115). Hence, the compression ratio for this scheme is expected to be

$$\rho_{\text{LLM+AC}} \approx \frac{\sum_{i=1}^{N_T} \log_2 \frac{1}{q_i(x_i)}}{\sum_{i=1}^{N_T} b_i}. \tag{3}$$

Values $\rho_{\text{LLM+zlib}}$, $\rho_{\text{LLM+TbyT}}$, and $\rho_{\text{LLM+AC}}$ provide upper bounds on $H(\mathbf{S})$.

## 2.2 ENTROPY BOUNDS

This section explores entropy bounds more rigorously. Let $\mathbf{S} \in \mathcal{S}^\infty$ be a random process that represents language input. The $n$th character in the sequence is denoted by $S_n$, whereas the string

of characters from the beginning to the $n$th character is expressed as $\mathbf{S}_n$. The tokenizer parses the input string and maps it to a sequence of tokens $\mathbf{X} = X_1, X_2, \ldots$ using a variable-length mapping. In this sequence, $X_i$ is the $i$th token. The number of characters employed to generate $X_i$ depends on the realization of the random process and, as such, we introduce random variable $B_i$ to identify the number of characters contained in the $i$th token. Motivated by practical considerations, we only admit tokenizers for which $B_i \geq 1$ and $B_i$ is uniformly bounded, with $B_i < \overline{B} < \infty$; these are characteristics of commonly used tokenizers. An immediate consequence of this framework is that, as the number of tokens grows unbounded $N_T \to \infty$, the number of characters must also approach infinity $N_c \to \infty$. Formally, consider the tokenizer function $T : \mathcal{S}^{\mathbb{N}} \to \mathcal{X}^{\mathbb{N}}$ operating on infinite symbol sequences; that is, $T(\mathbf{s}) = \mathbf{x}$ where $\mathbf{s}$ is an infinite sequence in $\mathcal{S}^{\infty}$. For natural number, $i \in \mathbb{N}$, define $m_i : \mathcal{S}^{\mathbb{N}} \to \mathbb{N}$ to be the (time) index during which the tokenizer working sequentially on an input sequence $\mathbf{s}$ outputs its $i$th token. Specifically, suppose $\mathbf{s}$ is given, then

$$m_i(\mathbf{s}) = \min_n \left\{ \text{length} \left( T(\mathbf{s}_n) \right) \geq i \right\}. \tag{4}$$

We note that, by construction, $\lim_{n\to\infty} \text{length} \left( T(\mathbf{s}_n) \right) = \infty$ and, as such, $m_i(\cdot)$ is well-defined. It may be pertinent to stress that the tokenizer function applied to truncated sequences is not necessarily injective because multiple finite input series can map to the same output. This phenomenon is a consequence of the fact that, at any point in time, a tokenizer working sequentially may be waiting for an additional symbol before it can unambiguously select the next output token, i.e., there may be instances where $T(\mathbf{s}_n) = T(\mathbf{s}_{n+1})$. However, if we restrict the input series to input indices when a new token is produced, then the restricted mapping becomes injective. That is, suppose $T(\mathbf{s}) = \mathbf{x}$, then the only (finite) series of input symbols in the restricted set for which $T(\mathbf{y}_n) = \mathbf{x}_i$ is $\mathbf{s}_{m_i(\mathbf{s})}$. Given a fixed sequence $\mathbf{s}$, we can express the number of characters contained in a token as

$$b_i = m_i(\mathbf{s}) - m_{i-1}(\mathbf{s})$$

with initial condition $m_{-1} = 0$. Consequently, the number of characters embedded in the first $N_T$ tokens for a random input becomes $N_c = \sum_{i=1}^{N_T} B_i$.

Having established these properties, we turn to the relation between $H(\mathbf{S})$ and $H(\mathbf{X})$. We make the assumption that $\{S_k\}_{k=1}^{\infty}$, $\{B_i\}_{i=1}^{\infty}$, and $\{X_i\}_{i=1}^{\infty}$ are stationary and ergodic processes. We know from the Shannon-McMillan-Breiman Theorem Cover & Thomas (1999) that

$$-\frac{1}{n} \log_2 p_{\mathbf{S}_n}(S_1, \ldots, S_n) = -\frac{1}{n} \log_2 p_{\mathbf{S}_n}(\mathbf{S}_n) \to H(\mathbf{S}) \quad \text{almost surely.} \tag{5}$$

Let $\Omega_{\mathbf{S}}$ be the collection of $\omega \in \Omega$ for which this limit holds. In an analogous manner, the Shannon-McMillan-Breiman theorem implies

$$-\frac{1}{i} \log_2 p_{\mathbf{X}_i}(X_1, \ldots, X_i) = -\frac{1}{i} \log_2 p_{\mathbf{X}_i}(\mathbf{X}_i) \to H(\mathbf{X}) \quad \text{almost surely.} \tag{6}$$

Define $\Omega_{\mathbf{X}}$ as the collection of $\omega \in \Omega$ for which this limit holds. Finally, by construction, we have

$$\lim_{i \to \infty} \frac{m_i(\mathbf{S})}{i} = \mathbb{E}[B] \quad \text{almost surely.} \tag{7}$$

Set $\Omega_B$ to be the set of $\omega \in \Omega$ for which this limit holds. For any $\omega \in \Omega_{\mathbf{S}} \cap \Omega_{\mathbf{X}} \cap \Omega_B$, we deduce that

$$\begin{aligned} H(\mathbf{S}) &= \lim_{k \to \infty} -\frac{1}{k} \log_2 p_{\mathbf{S}_k}(\mathbf{S}_k(\omega)) \\ &= \lim_{i \to \infty} -\frac{1}{l_i} \log_2 p_{\mathbf{S}_{l_i}}(\mathbf{S}_{l_i}(\omega)) \\ &= \lim_{i \to \infty} -\frac{1}{l_i} \log_2 \Pr \left( \mathbf{X}_i = T(\mathbf{S}_{l_i}(\omega)) \right) \\ &= -\frac{1}{\mathbb{E}[B]} \lim_{i \to \infty} \frac{1}{i} \log_2 \Pr \left( \mathbf{X}_i = \mathbf{x}_i \right) = \frac{H(\mathbf{X})}{\mathbb{E}[B]}. \end{aligned}$$

The first equality follows from equation 5. The second equality is a consequence of the fact that $\{l_i = m_i(\mathbf{S}(\omega)) | i \in \mathbb{N}\}$ is an infinite subset of the natural numbers. Since a subsequence of a convergent sequence must converge to the same limit, we immediately gather that this alternate form

approaches $H(\mathbf{S})$. The third equality is a consequence of the equivalence between the following two events,

$$\{\omega \in \Omega | \mathbf{X}_i(\omega) = \mathbf{x}_i\} = \{\omega \in \Omega | T(\mathbf{S}_{m_i(\mathbf{S}(\omega))}) = \mathbf{x}_i\}.$$

This is characteristic of the tokenization process, and it is a consequence of the correspondence described above. The last step holds because we are considering an $\omega \in \Omega_B$. The sets $\Omega_{\mathbf{S}}$, $\Omega_{\mathbf{X}}$, and $\Omega_B$ each have probability one; this implies that their intersection also has probability one, Thus, we must conclude that

$$H(\mathbf{S}) = \frac{H(\mathbf{X})}{\mathbb{E}[B]} \quad \text{almost surely.}$$

As a corollary to this result, any upper bound on $H(\mathbf{X})$ produces an upper bound on $H(\mathbf{S})$. This is the property we wish to exploit.

Then, from the results of Cover & King (1978), we can see that

$$\Pr\left\{H(\mathbf{X}) \leq \lim_{N_T \to \infty} -\frac{1}{N_T} \sum_{i=1}^{N_T} \log_2 q_i(X_i)\right\} = 1, \tag{8}$$

where $q_i(\cdot)$ is the output PMF from the language model. Therefore, an asymptotic upper bound on the entropy rate $H(\mathbf{S})$ is given by

$$H(\mathbf{S}) \leq \frac{\lim_{N_T \to \infty} -\frac{1}{N_T} \sum_{i=1}^{N_T} \log_2 q_i(X_i)}{\mathbb{E}[B]}. \tag{9}$$

We refer to the expression in the right-hand side of equation 9 as the asymptotic upper bound on $H(\mathbf{S})$ and denote it by $H_{ub}$. The numerator in equation 9 represents the average number of bits required to represent the tokens $\mathbf{X}_{N_T}$, and the denominator in equation 9 is the average number of characters per token. Hence, the unit for $H(\mathbf{S})$ is bits/character. In Cover & King (1978), the authors offer 1.3 bits/character as an estimate of the asymptotic upper bound on $H(\mathbf{S})$. They also provide an extensive list of references and a discussion of the literature on estimating the entropy of English prior to 1976. Very recently, in (Lutati et al., 2023, Table 4), the performance of several language models have been evaluated on the text8 dataset using a metric called bits per character (bpc). The bpc metric in Lutati et al. (2023) is the same as the asymptotic upper bound in this paper.

It is important to emphasize that $H_{ub}$, $\rho_{\text{LLaMA+zlib}}$, $\rho_{\text{LLaMA+TbyT}}$, and $\rho_{\text{LLM+AC}}$ are estimated using a finite number of tokens; the statistical properties of such estimates should be kept in mind when interpreting the results, especially since the tokens are from a very large alphabet and the language model has a large memory.

## 3 LLM-BASED COMPRESSION WITH SIDE INFORMATION

In this section, we discuss the value of side information in the form of a summary of the text, when it is available either during compression alone, or both during compression and decoding.

### 3.1 SUMMARY AVAILABLE AT THE ENCODER

We first consider the case where the text is compressed by conditioning on a summary, which is generated at the encoder. Let $\mathbf{S}$ denote the text to be compressed and let $g(\mathbf{S})$ denote a function of the text such as, for example, a summary. The joint entropy of $\mathbf{S}$ and $g(\mathbf{S})$ can be decomposed in two different ways,

$$H(\mathbf{S}, g(\mathbf{S})) = H(\mathbf{S}) + H(g(\mathbf{S})|\mathbf{S}) = H(g(\mathbf{S})) + H(\mathbf{S}|g(\mathbf{S})) \tag{10}$$
$$\implies H(\mathbf{S}) = H(g(\mathbf{S})) + H(\mathbf{S}|g(\mathbf{S})). \tag{11}$$

Above, equation 11 follows from equation 10 because $g(\cdot)$ is a deterministic function and, hence, $H(g(\mathbf{S})|\mathbf{S}) = 0$. Intuitively, the implication of equation 11 is that it is (nearly) optimal to compress $\mathbf{S}$ by using $H(g(\mathbf{S}))$ bits to compress $g(\mathbf{S})$ and using $H(\mathbf{S}|g(\mathbf{S}))$ bits to compress $\mathbf{S}$ by conditioning on $g(\mathbf{S})$. An optimal universal compression algorithm may implicitly leverage these notions and, when such an algorithm is available, there is no discernible benefit to explicitly taking this two-step

approach. Yet, LLM-based compressors are not necessarily universally optimal – particularly for finite context window lengths and, therefore, their gap to optimality may be different depending on whether we compress $\mathbf{S}$ with or without conditioning on $g(\mathbf{S})$. Although the English language is often modeled as an ergodic random process, a finite length of English text may exhibit substantial variations in its statistical properties based on the underlying context. In this case, $g(\mathbf{S})$ can act as the hidden context, and, hence, the two-step approach can aid in the compression of $\mathbf{S}$.

We perform compression using the summary $g(\mathbf{S})$ as follows. The summary $g(\mathbf{S})$ is parsed into a sequence of tokens denoted by $\mathbf{y}$. We first use LLM+AC to compress $g(\mathbf{S})$ and obtain $N_g$ compressed bits for the summary. Then, the LLM+AC algorithm is adapted to compress the text $\mathbf{S}$ such that at each epoch $i$, we pass the summary tokens $\mathbf{y}$, along with the past $M$ tokens to the LLM, to obtain a conditional PMF $\tilde{q}_i(x_i) := \Pr(\mathbf{X}_i = x_i | \mathbf{y}, x_{i-1}, x_{i-2}, \ldots, x_{i-M}), \forall x_i \in \mathcal{X}$. The conditional PMF vector $\tilde{\mathbf{q}}_i$ is then passed to the arithmetic coder, which produces a sequence of $N_b$ bits. At the decoder, the summary is first decoded and then added to the context window similar to what is done at the encoder. The compression ratio ($\tilde{\rho}$) of this scheme is $\tilde{\rho} = \frac{N_g + N_b}{N_c}$, where $N_c$ is the number of characters in the text $\mathbf{s}$ alone.

It should be noted that compressing $g(\mathbf{S})$ requires an additional $N_g \geq H(g(\mathbf{S}))$ bits. Therefore, the gain in compressing $\mathbf{S}$ by conditioning on $g(\mathbf{S})$ should exceed $N_g$ bits for the two-step approach to be beneficial. In Section 4, we show that this is indeed the case in some cases.

## 3.2 Summary available at both the encoder and decoder

We next consider the compression performance of LLM+AC when $g(\mathbf{S})$ is available as side information to both the encoder and the decoder. Since $H(\mathbf{S}|g(\mathbf{S})) \leq H(\mathbf{S})$, we expect an improvement in the compression performance if LLM+AC can exploit the side information, . As in Sec. 3.1, the side information is added to the beginning of the context window for all $i$. The difference from Sec. 3.1 lies in the fact that $g(\mathbf{S})$ does not have to be compressed since it is available at the decoder.

## 4 Results

We used the two versions of the LLaMA foundation models, which are LLaMA-7B Touvron et al. (2023a) and LLaMA2-7B Touvron et al. (2023b) as the large language models for our results. We will refer to them as LLaMA and LLaMA2, respectively.

It should be noted that the tokenizer and the model are trained on a large corpus of text which includes uppercase letters, special characters, etc. This is in contrast to many studies on estimating the entropy of English, where the input alphabet is restricted to lowercase letters such as in Shannon (1951); Cover & King (1978); Goyal et al. (2018). This makes it difficult to perform an entirely fair comparison between these models. By using a pre-trained LLM on an input consisting only of lowercase letters, we may be unfair to the LLM. We will show that LLM-based compression has excellent performance regardless of this issue.

## 4.1 Compression performance when using LLaMA models on text8 dataset

In our first experiment, we used the text8 dataset available from `http://mattmahoney.net/dc/text8.zip` to benchmark the performance of LLaMA2 with compression against other state-of-the-art results for text compression. In Goyal et al. (2018), it is mentioned that the ZPAQ algorithm obtains the best compression ratio for the text8 dataset with a compression ratio of 1.4 bits/character. In Mahoney (2011), the paq8h algorithm provides a compression ratio of 1.2 bits/character. To the best of our knowledge, this appears to be the best performance reported. Therefore, we used these two algorithms as baselines. We did not independently run the ZPAQ or paq8h algorithms; we are quoting results from the existing literature.

The performance of LLaMA2 is shown in Table 1 for various memory($M$) lengths. It can be seen that using LLaMA2 with Arithmetic Coding and a memory of 511 results in a compression ratio of 0.6936 bits/character. This is substantially better than the state-of-the-art results mentioned in Goyal et al. (2018) or Mahoney (2011) and is very close to our computed upper bound. The performance with the LLaMA2+zlib algorithm and LLaMA2+TbyT compression is also better than that of the

known state-of-the-art results. Table 1 also shows the upper bound in equation 9. It should be noted that the upper bound on the entropy is lower than that computed by Shannon in Shannon (1951), Cover and King in Cover & King (1978) and more recent estimates based on neural networks in Lutati et al. (2023). It can be seen that the compression performance improves with increasing $M$.

We included the performance of LLaMA+AC in the last column, and we can observe that using LLaMA2 indeed improves the performance across all memory lengths. We also observed that the inference time scaled approximately linearly with the input memory length, i.e., batches with a memory of 511 tokens ran about 16 times slower than batches with a memory of 31 tokens.

Table 1: Compression performance of LLMZip on text8 dataset as a function of its memory ($M$)

| $M$ | $N_c$ | $N_t$ | $H_{ub}$ (bpc) | $\rho_{LLaMA2+zlib}$ file size (bits) | $\rho_{LLaMA2+TbyT}$ (bpc) | $\rho_{LLaMA2+AC}$ (bpc) | $\rho_{LLaMA+AC}$ (bpc) |
|---|---|---|---|---|---|---|---|
| 31 | $4,568,855$ | $1,000,000$ | 0.8955 | 1.2963 | 1.0241 | **0.8962** | 0.9145 |
| 127 | $4,568,855$ | $1,000,000$ | 0.7343 | 1.1104 | 0.8685 | **0.7352** | 0.752 |
| 255 | $4,568,855$ | $1,000,000$ | 0.708 | 1.0792 | 0.8435 | **0.7089** | 0.725 |
| 511 | $4,568,855$ | $1,000,000$ | 0.6927 | 1.061 | 0.8292 | **0.6936** | 0.7101 |

Note that we did not run the LLaMA models on the entire 100MB of the text8 dataset due to compute limitations. Hence, the comparison with the state-of-the-art corresponds to estimates obtained from different input sizes. Also, since the text8 dataset is derived from Wikipedia on which LLaMA models were trained, it is likely that our results for the text8 dataset are optimistic. We next test on a problem for which we are certain that the LLaMA models were not trained.

## 4.2 Compression performance when using LLaMA2 on a recent book

We study the performance of LLaMA2 on a recently released (Sep 20, 2023) book Cooke & Reed (2023) under Project Gutenberg, which appeared after LLaMA2 was published. We extracted text that corresponds to 50,000 tokens. We applied the same text pre-processing as used in the text8 dataset to clean the text from the book. The resulting text data contained only lowercase letters and space as in the text8 dataset. Table 2 shows the compression performance of the LLM on the book. It can be seen that the compression ratios and the entropy upper bound are slightly higher compared to the performance on the text8 dataset; nevertheless, the asymptotic upper bound on the entropy is lower than that of currently known models given in (Lutati et al., 2023, Table 4)). Similarly, the compression ratios of LLaMA2-based compressors are better than those of known state-of-the-art results for the text8 dataset. The compression ratio for LLaMA2 with arithmetic coding is only 0.7741 bits/character and is very close to the estimated upper bound on $H(\mathbf{S})$.

Table 2: Compression performance of the LLaMa2 + AC on a recently published book in Project Gutenberg Cooke & Reed (2023), as a function of its memory ($M$)

| $M$ | $N_c$ | $N_t$ | $H_{ub}$ (bpc) | $\rho_{LLaMA2+zlib}$ (bpc) | $\rho_{LLaMA2+TbyT}$ (bpc) | $\rho_{LLaMA2+AC}$ (bpc) | Standalone Zlib (bpc) |
|---|---|---|---|---|---|---|---|
| 31 | $210,682$ | $48,921$ | 1.0974 | 1.5571 | 1.2296 | 1.0977 | 2.55 |
| 127 | $210,682$ | $48,921$ | 0.8544 | 1.2783 | 0.9941 | 0.8552 | 2.55 |
| 255 | $210,682$ | $48,921$ | 0.8047 | 1.2213 | 0.9474 | 0.8054 | 2.55 |
| 511 | $210,682$ | $48,921$ | 0.7733 | 1.1826 | 0.9172 | 0.7741 | 2.55 |

To provide some insight into the comparative performance of LLaMA2 based compressors vis-a-vis standard text compressors, we also ran the zlib algorithm directly on the input text. The resulting compression ratio was 2.55 bits/character (shown in the last column). It is clear that the performance of LLaMA2 based compressors is substantially better than this. The zlib algorithm

may not be optimized for compressing small text samples and hence, the compression ratio for the zlib algorithm and the LLaMA2+zlib will likely improve on longer texts.

### 4.3 COMPRESSION PERFORMANCE OF LLAMA2 MODELS ON A RESEARCH PAPER BY USING A SUMMARY OF THE PAPER

In Table 3, we report the compression performance of LLaMa2 + AC on a recently published research paper Li et al. (2023) in arXiv. We evaluate six different texts as summaries which include the paper's abstract, the first paragraph in the introduction, and four summaries of varying lengths that were generated using GPT-4. We compress the Latex source code available for the paper and do not include the abstract as part of the compression. We observe that when the summary is available as side information at both the encoder and decoder, the compression ratio ($\rho_{LLaMa2+AC}$), always improves. It's worth noting that Summary 3 and Summary 4 offer the maximum improvements compared to others, and yet Summary 3 is roughly half the size of that of Summary 4. We further noted that Summary 3 is less compressible than Summary 2, which suggests that it is more informative. It is also worth noting that Paragraph 1 offers the least improvement, indicating the improvements are indeed due to the conditioning of a valid summary, and conditioning on just a paragraph may not yield significant improvements.

For the case when the summary is only available at the encoder, we only get a marginal improvement in the compression ratio denoted by $\tilde{\rho}_{LLaMa2+AC}$. Summary 3 offers the best compression ratio, while Summary 4 and Paragraph 1 have compression ratios worse than the no summary case.

Conditioning on summaries is much less computationally expensive than using an LLM with a larger memory $M$. This is because the summaries are fixed text, at a fixed position and hence the corresponding keys, queries, and values can be computed once and reused for the subsequent attention mechanisms in each epoch. This is not the case when the LLM has larger $M$, as for every stride across the text, the positional encodings change and hence the keys and values of the tokens present in previous epochs cannot be reused.

Table 3: Compression performance of LLaMa2 +AC with summary and M = 511 on a recently published paper in arXiv Li et al. (2023)

| Type | Summary | | | Text | | | $N_g + N_b$ | $\rho_{LLaMa2+AC}$ | $\tilde{\rho}_{LLaMa2+AC}$ |
| --- | --- | --- | --- | --- | --- | --- | --- | --- | --- |
| | $N_{tg}$ | $N_{cg}$ | $N_g$ | $N_t$ | $N_c$ | $N_b$ | | (bpc) | (bpc) |
| No Summary | 0 | 0 | 0 | 12578 | 43787 | 26354 | 26354 | 0.6019 | 0.6019 |
| Abstract | 252 | 1312 | 662 | 12578 | 43787 | 25596 | 26258 | 0.5846 | 0.5997 |
| Summary 1 | 219 | 1095 | 611 | 12578 | 43787 | 25612 | 26223 | 0.5849 | 0.5989 |
| Summary 2 | 473 | 2233 | 1281 | 12578 | 43787 | 24836 | 26117 | 0.5672 | 0.5965 |
| Summary 3 | 428 | 1918 | 1375 | 12578 | 43787 | 24370 | 25745 | 0.5566 | 0.5880 |
| Summary 4 | 945 | 4919 | 2228 | 12578 | 43787 | 24328 | 26556 | 0.5556 | 0.6064 |
| Paragraph 1 | 281 | 1255 | 931 | 12578 | 43787 | 26190 | 27121 | 0.5981 | 0.6194 |

## 5 CONCLUSION

In this paper, we introduced a suite of lossless text compression algorithms that use an LLM (LLaMA2-7B) in conjunction with entropy coding. The LLM is used to predict the next token given a window of past tokens as memory. Our suite of algorithms outperforms state-of-the-art compression algorithms like BSC, ZPAQ and paq8h. We obtained our best result when using LLaMA2-7B with arithmetic coding and a memory of 512 tokens. We also observed that LLMs with greater memory length showed better performance. Additionally, we provided estimates of an asymptotic upper bound on the entropy of English, which are significantly smaller than those found in prior work. Finally, we showed that when using LLaMA2 + AC for compressing a document, a marginal performance improvement can be obtained by first extracting a summary of the document, and compressing the document by conditioning on this summary. We also show that when such a summary is available at both the encoder and decoder a significant performance improvement can be obtained.

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
