# OpenReview forum: "LLMZip: Lossless Text Compression using Large Language Models"
_ICLR.cc/2024/Conference — Submitted to ICLR 2024_

### Official Review · Reviewer_o5QE · 2023-10-28

**Soundness:** 2 fair
**Presentation:** 2 fair
**Contribution:** 2 fair
**Rating:** 6
**Confidence:** 3

**Summary:**

This paper presents LLMZip, an algorithm that leverages the recently developed Transformer-decoder-based Large Language Models (LLMs) for better compression. In this paper, we study the combination of LLM with various lossless compression schemes and analyze their empirical performance. We also explore how LLMZip's performance can be further enhanced with the availability of side information, such as text summaries. Experimental studies on various datasets are provided to demonstrate the effectiveness of the proposed LLMZip approach, which indicates that LLMZip achieves a better bits-per-character ratio compared to the traditional Zlib lossless compression method.

**Strengths:**

- The proposed idea is intuitive and easy to follow.
- Exploring the potential applications of LLMs in compression is a promising research direction.
- The paper is well presented in general.

**Weaknesses:**

- The novelty of the LLMZip method seems to be limited. As mentioned in the introduction, it seems the connection between compression and prediction has been developed decades ago. Using language models for compression has also been explored before, with LSTM and/or RNN being used. Therefore, it appears that the sole contribution of this paper is the substitution of previously explored smaller language models with large language models (LLMs)
- Only Llama2 has been used as the language model in the experiment. It is unclear how the performance of LLMZip would vary when employing LLMs of different types and sizes.
- LLMs usually require GPUs for execution, whereas traditional compression algorithms can run on CPUs, which are more widely accessible and easier to democratize.
- The LLMZip approach presented in this paper appears to be more suitable for submission to an information theory conference or journal, such as ISIT.

**Questions:**

- How does the performance of LLMZip vary when using aligned LLMs, e.g., Vicuna or fine-tuned Llama2?
- How does the performance of LLMZip vary when using LLMs with various scales, e.g., 13B or 65/70B?
- What does the end-to-end running time of LLMZip look like when compared to Zlib?

---

> ### Author Response · Authors · 2023-11-22
>
> Dear Reviewer,
> We appreciate your time and effort in reviewing our work. We would like to address each point you have raised in the review.
>
> > 1. The novelty of the LLMZip method seems to be limited. As mentioned in the introduction, it seems the connection between compression and prediction has been developed decades ago. Using language models for compression has also been explored before, with LSTM and/or RNN being used. Therefore, it appears that the sole contribution of this paper is the substitution of previously explored smaller language models with large language models (LLMs)
>
> Prior to our research, the compression performance of pre-trained LLMs with Arithmetic Coding had not been reported. While there have been works that utilized LSTMs (Deepzip) and Transformers (NNCPv2) for text compression, LLaMA1 and LLaMA2 significantly outperform earlier results using LSTMs and Transformers. We believe it is necessary to bring this to the attention of the ML community since this may have important consequences for the practice of compression in some scenarios. The idea of using the summary as side information is new.
>
> > 2. Only Llama2 has been used as the language model in the experiment. It is unclear how the performance of LLMZip would vary when employing LLMs of different types and sizes.
>
> We also report the performance of using LLaMa1, which is the older model, in Table 1, and we do see an improvement in performance when LLaMa2 is used. Using LLMs out of the box for LLMzip compression can be very computationally intensive. As the experiments involve processing 1M tokens for each context size , more gpus are needed for longer durations when using larger versions. Hence we limited our focus to just the 7B version of LLaMa.
>
> > 3. LLMs usually require GPUs for execution, whereas traditional compression algorithms can run on CPUs, which are more widely accessible and easier to democratize.
>
> This is a valid point. Yet, the article points to a significant gap and a new approach warrants dissemination. The approach may not be ready for widespread dissemination; however, it is a significant step in a new direction.
>
> > 4. The LLMZip approach presented in this paper appears to be more suitable for submission to an information theory conference or journal, such as ISIT.
>
> The aim of this work is to disseminate this information to the ML / LLM community that LLMs can do SOTA compression. We believe it is important to disseminate this to the ML community rather than the IT community, because additional work is required from the ML community to make these practical, and this may have a significant impact on the future of compression in many scenarios. As quantization and other techniques to accelerate LLM inference become better, we believe this complexity can be a lot more manageable. There is scope to engineer better solutions.
>
> > 5. How does the performance of LLMZip vary when using aligned LLMs, e.g., Vicuna or fine-tuned Llama2?
> 6. How does the performance of LLMZip vary when using LLMs with various scales, e.g., 13B or 65/70B?
>
> We haven’t tried LLMZip with LLMs other than LLaMa1 and LLaMa2, as we believe they suffice to show that pre-trained LLMs do indeed offer substantial compression improvements.
>
> > 7. What does the end-to-end running time of LLMZip look like when compared to Zlib?
>
> The run times for 100K tokens is 10 hrs for 512 context length and  35 mins for a context length of 32, while zlib does it in less than a second. While these are clearly not impressive run times, these are just out-of-the-box applications, with a lot of scope to be improved. It is also worth noting that 32 context length LLM still significantly outperforms Zlib compression. We do not have runtime comparisons for the other SOTA algorithms like paq8h or bsc.
>
> We hope this addresses your concerns and we look forward to your further feedback.

---

### Official Review · Reviewer_1ivF · 2023-10-29

**Soundness:** 3 good
**Presentation:** 3 good
**Contribution:** 3 good
**Rating:** 6
**Confidence:** 3

**Summary:**

This paper proposes a LLM based text compression algorithm and empirically demonstrate the superiority of the proposed method over commonly used compression approaches. The authors also explores the usage of side information such as summary of the text to boots the compression performance further.

**Strengths:**

- The authors improve the text compression performance by combining LLMs and arithmetic coding.
- The paper provides new estimates of an asymptotic upper bound on the entropy of English

**Weaknesses:**

- The authors should provide more background introduction about text compression and make the paper more self-contained.

**Questions:**

- What are the baseline results mentioned in Table 1? I would suggest the authors include more introductions about the baseline methods and their performances.

- Table 3 shows the performance varies based on the summary quality. How can we determine if the side information is helpful or not before compression?

- Why does side information available in the encoder and decoder perform better than the encoder-only?

- From the perspective of evaluation metric, does a lower bpc indicate a stronger LLM model potential in common downstream NLP tasks(summarization, translation, classification, etc.)?

---

> ### Author Response · Authors · 2023-11-22
>
> Dear Reviewer,
>
> Thank you for your valuable feedback and insightful queries regarding our work. We would like to address each point you have raised in the review.
>
> > 1. What are the baseline results mentioned in Table 1? I would suggest the authors include more introductions about the baseline methods and their performances.
>
> In Table 1, we presented results for different encoding schemes which have been explained in sections 2.1.1, 2.1.2, and 2.1.3, and the entropy bound was presented in section 2.2. We compare our results to ZPAQ and PAQ8h in the discussion of the results (section 4.1). These are state-of-the-art algorithms as reported in various papers on compression. Given the page limitations, we weren’t able to include more background on these algorithms, but we can include more references to these.
>
> > 2. Table 3 shows the performance varies based on the summary quality. How can we determine if the side information is helpful or not before compression?
>
> Thank you for your insightful question about the variability of performance based on summary quality. While it is challenging to precisely quantify the effectiveness of a summary prior to compression, our observations indicate a trend. Particularly, a detailed and less compressible summary (like Summary 3 in our study) tends to offer more valuable information for compression, as seen in its high compression ratio (1918 characters to 1375 bits, or 0.72 bits per character).
>
> > 3. Why does side information available in the encoder and decoder perform better than the encoder-only?
>
> When side information is available only at the encoder, the compression ratio is computed as (compressed bits of summary + compressed bits of text)/(number of characters in text), denoted by $\tilde{\rho}$​. When the summary is available at both ends, the compressed summary bits need not be transmitted/stored, hence we need not account for it in the compression ratio.
>
> > 4. From the perspective of evaluation metric, does a lower bpc indicate a stronger LLM model potential in common downstream NLP tasks (summarization, translation, classification, etc.)?
>
> We are not entirely sure whether a lower bpc indicates a stronger LLM model potential in downstream NLP tasks. We do not make any such claims in the paper. We also don’t think this possibility can be ruled out either given the consistent performance improvement of LLaMA2 over LLaMA1 (as shown in Table 1, last two columns) across all window sizes.
>
> We hope this addresses your concerns and we look forward to your further feedback.

---

### Official Review · Reviewer_fbKB · 2023-10-30

**Soundness:** 2 fair
**Presentation:** 1 poor
**Contribution:** 1 poor
**Rating:** 3
**Confidence:** 4

**Summary:**

This paper examines text compression utilizing the pretrained Large Language Model (LLaMa2) in conjunction with Arithmetic Coding (AC). The authors present an estimation of the entropy rate of the input text. Additionally, by using a summary as side information, they achieve a marginally improved compression ratio.

**Strengths:**

Utilizing a state-of-the-art Large Language Model for text compression to attain a higher compression ratio is interesting.

**Weaknesses:**

1. The concept of compressing text with pretrained language models is not groundbreaking. The paper's attempt to innovate using the advanced LLaMa2 as an LLM seems to lack strong novelty.

2. Sections 2.1.1 and 2.1.2, which discuss text compression methods other than arithmetic coding, appear superfluous.

3. The detailed explanation of Arithmetic coding might be redundant; perhaps it would be better placed in an appendix.

4. In Section 2.2 regarding Entropy bounds, the equation H(S) = H(X) / E[B] does not appear to be a significant finding.

5. The use of text-summary for compression seems misjudged. Theoretically, adding bits to describe a summary would only be beneficial if the probability estimation isn't flawless.

6. The claims about the entropy bounds of the English language are debatable. For instance, Table 1 lists 0.6936, while Table 2 cites 0.7741 from a different dataset. The input data chosen for testing does not seem to be a true representation of English text, with entropy rates that fluctuate depending on the input.

7. Minor Remarks:
- The use of "It's" may not be suitable for a formal paper.
- Terminologies such as $N_{tg}$, $N_{cg}$, and others need clear and precise definitions.

**Questions:**

Please see Weakness section.

---

> ### Author Response · Authors · 2023-11-22
>
> Dear Reviewer,
>
> We are grateful for the time and effort you have invested in reviewing our work. However, we believe the reviewer has overlooked the main contributions of the paper and the significance of the results.
>
> > 1. The concept of compressing text with pre-trained language models is not groundbreaking. The paper's attempt to innovate using the advanced LLaMa2 as an LLM seems to lack strong novelty.
>
> “Groundbreaking” is a relative term and we cannot argue with the reviewer’s view of what is groundbreaking. There are two important results in this paper that need to be disseminated to the research community. First, we emphasize that the compression performance of pre-trained Large Language Models  (LLMs) with Arithmetic Coding has not been previously explored. While there are studies employing LSTMs (e.g., Deepzip) and Transformers (e.g., NNCPv2) for text compression, their performance does not match the substantial results achieved using LLaMA1 or LLaMA2 for compression. The gap is not small and we believe this will have a significant influence on the future of compression, at least in some scenarios. Secondly, the idea of using a summary as side information to compress the text is novel (please see our response to Question 4 for more details).
>
> > 2. Sections 2.1.1 and 2.1.2, which discuss text compression methods other than arithmetic coding, appear superfluous. The detailed explanation of Arithmetic coding might be redundant; perhaps it would be better placed in an appendix.
>
> We included a detailed explanation of Arithmetic coding in our paper to ensure it is self-contained and accessible to those unfamiliar with the concept. This decision aligns with the suggestion from reviewer 1ivf to provide more background on text compression. We believe our approach achieves a balance, but we are open to relocating this section to an appendix if you find it more appropriate.
>
> > 3. In Section 2.2 regarding Entropy bounds, the equation H(S) = H(X) / E[B] does not appear to be a significant finding.
>
> Section 2.2 is necessary to formally examine entropy bounds in the presence of tokenization in terms of bits per character. In particular, it is important to realize that the token sequence has to satisfy the conditions necessary for the Shannon McMillan Breiman theorem to hold for us to meaningfully use these entropy bounds as a lower bound for compression in terms of bits/character.
>
> > 4. The use of text summary for compression seems misjudged. Theoretically, adding bits to describe a summary would only be beneficial if the probability estimation isn't flawless.
>
> We are not sure why the reviewer thinks that this section is misjudged. In reality, every language model only provides a flawed estimate of the conditional probability and hence, this section is very much relevant to reality. This is an idea that, to the best of our knowledge, hasn’t been presented in any other work. We have provided empirical evidence that this is beneficial when used with a real LLM (Llama2).
>
> > 5. The claims about the entropy bounds of the English language are debatable. For instance, Table 1 lists 0.6936, while Table 2 cites 0.7741 from a different dataset. The input data chosen for testing does not seem to be a true representation of English text, with entropy rates that fluctuate depending on the input.
>
> There is nothing called a “true representation of English text” that can be used for testing. The statistical nature of the English language is complicated and estimates of the bounds will vary with the realization. Our estimates of the bound (0.6936 / 0.7741) show that the upper bound estimate may be significantly lower than the estimates from Thomas & Cover (1.2) and the reviewer should focus on how large this difference is and the implications of this. Of course, we can average the entropy over more and more inputs to get sharper estimates. That would be a pedantic exercise.
>
> We hope that this response addresses your concerns and clarifies the contributions and intentions of our work. We remain open to further discussion and are willing to make modifications to our paper to better reflect the valuable feedback provided.

---

### Meta-Review · Area_Chair_EZuq · 2023-12-11

**Metareview:**

This paper presents LLMZip algorithm that uses arithmetic coding and LLM to do text compression. This is not surprising as the conditional probabilities produced by the LLM can be very informative. The authors also presents a new estimate of asymptotic upper bound for English.

Strength:
1. An important application of LLM for text compression.
2. A new estimate of asymptotic upper bound for English.

Weakness:
1. Novelty is quite limited as it is well expected that LLM can provide very accurate predictive probabilities.
2. It is computationally very expensive to do for the purpose of text compression using large language models. Unless it is significantly better than other approaches, the impact is limited as well.

**Justification For Why Not Higher Score:**

See weakness.

**Justification For Why Not Lower Score:**

N/A

---

### Decision · Program_Chairs · 2024-01-16

Reject